

# Ensemble of adapted convolutional neural networks (CNN) methods for classifying colon histopathological images

Dheeb Albashish

Computer Science Department/ Prince Abdullah bin Ghazi Faculty of Information and Communication Technology, Al-Balqa Applied University, Alsalt, Jordan

## ABSTRACT

Deep convolutional neural networks (CNN) manifest the potential for computer-aided diagnosis systems (CADs) by learning features directly from images rather than using traditional feature extraction methods. Nevertheless, due to the limited sample sizes and heterogeneity in tumor presentation in medical images, CNN models suffer from training issues, including training from scratch, which leads to overfitting. Alternatively, a pre-trained neural network's transfer learning (TL) is used to derive tumor knowledge from medical image datasets using CNN that were designed for non-medical activations, alleviating the need for large datasets. This study proposes two ensemble learning techniques: E-CNN (product rule) and E-CNN (majority voting). These techniques are based on the adaptation of the pretrained CNN models to classify colon cancer histopathology images into various classes. In these ensembles, the individuals are, initially, constructed by adapting pretrained DenseNet121, MobileNetV2, InceptionV3, and VGG16 models. The adaptation of these models is based on a block-wise fine-tuning policy, in which a set of dense and dropout layers of these pretrained models is joined to explore the variation in the histology images. Then, the models' decisions are fused *via* product rule and majority voting aggregation methods. The proposed model was validated against the standard pretrained models and the most recent works on two publicly available benchmark colon histopathological image datasets: Stoean (357 images) and Kather colorectal histology (5,000 images). The results were 97.20% and 91.28% accurate, respectively. The achieved results outperformed the state-of-the-art studies and confirmed that the proposed E-CNNs could be extended to be used in various medical image applications.

# INTRODUCTION

Colon cancer is the third most deadly disease in males and the second most hazardous in females. According to the World Cancer Research Fund International, over 1.8 million new cases were reported in 2018 (*Belciug & Gorunescu, 2020*). In the diagnosis of colon cancer, the study of histopathological images under the microscope plays a significant role

Corresponding author
Dheeb Albashish, bashish@bau.edu.jo

in the interpretation of specific biological activities. Among the microscopic inspection functions, classification of images (organs, tissues, etc.) is one of considerable tasks. However, classifying medical images into a set of different classes is a very challenging issue due to low inter-class distance and high intra-class variability (*Sahran et al., 2018*), as illustrated in Fig. 1. Some objects in medical images may be found in images belonging to different classes, and different objects may appear at different orientations and scales in a given class. During the manual assessment, physicians examine the hematoxylin and eosin (H&E) stained tissues under a microscope to analyze their histopathological attributes, such as cytoplasm, nuclei, gland, and lumen, as well as change in the benign structure of the tissues. It is worth noting that early categorization of colon samples as benign or malignant, or discriminating between different malignant grades is critical for selecting the best treatment protocol. Nevertheless, manually diagnosing colon H&E stained tissue under a microscope is time-consuming and tedious, as illustrated in Fig. 1. In addition, the diagnostic performance depends on the experience and personal skills of a pathologist. It also suffers from inter-observer variability with around 75% diagnostic agreement across pathologists (*Elmore et al., 2015*). As a result, the treatment protocol might differ from one pathologist to another. These issues motivate development and research into the automation of diagnostic and prognosis procedures (*Stoean et al., 2016*).

In recent decades, various computer aided diagnosis systems (CADs) have been introduced to tackle the classification problems in cancer digital pathology diagnosis to achieve reproducible and rapid results (*Bicakci et al., 2020*). CADs assist in enhancing the classification performance and, at the same time, minimize the variability in interpretations (*Rahman et al., 2021*). The faults produced by CADs/machine learning model have been announced to be less than those produced by a pathologist (*Kumar et al., 2020*). These models can also assist clinicians in detecting cancerous tissue in colon tissue images. As a result, researchers are trying to construct CADs to improve diagnostic effectiveness and raise inter-observer satisfaction (*Tang et al., 2009*). Numerous conventional CADs for identifying colon cancer using histological images had been introduced by number of researchers in the past years (*Stoean et al., 2016*; *Kalkan et al., 2012*; *Li et al., 2019*). Most of the conventional CADs focus on discriminating between benign and malignant tissues. Furthermore, they focus on conventional machine learning and image processing techniques. In this regards, they emphasize on some complex tasks such as extracting features from medical images and require extensive preprocessing. The complex nature of these tasks in machine learning techniques degrades the results of the CADs regarding accuracy and efficiency (*Ahmad, Farooq & Ghani, 2021*). Conversely, recent advances in machine learning technologies make this task more accurate and cost-effective than traditional models (*Abu Khurma et al., 2022*; *Khurma, Aljarah & Sharieh, 2021*; *Abu Khurmaa, Aljarah & Sharieh, 2021*).

In the last few years, deep learning techniques have become a prevalent and leading tool in the field of machine learning for colon histopathlogical image classification. Recently, one of the most successful deep learning techniques is the deep convolutional neural networks (CNN) (*Khan et al., 2020*) that consists of series of convolutional and pooling layers. These are followed by the fully-connected (FCC) and softmax layers. The FCC

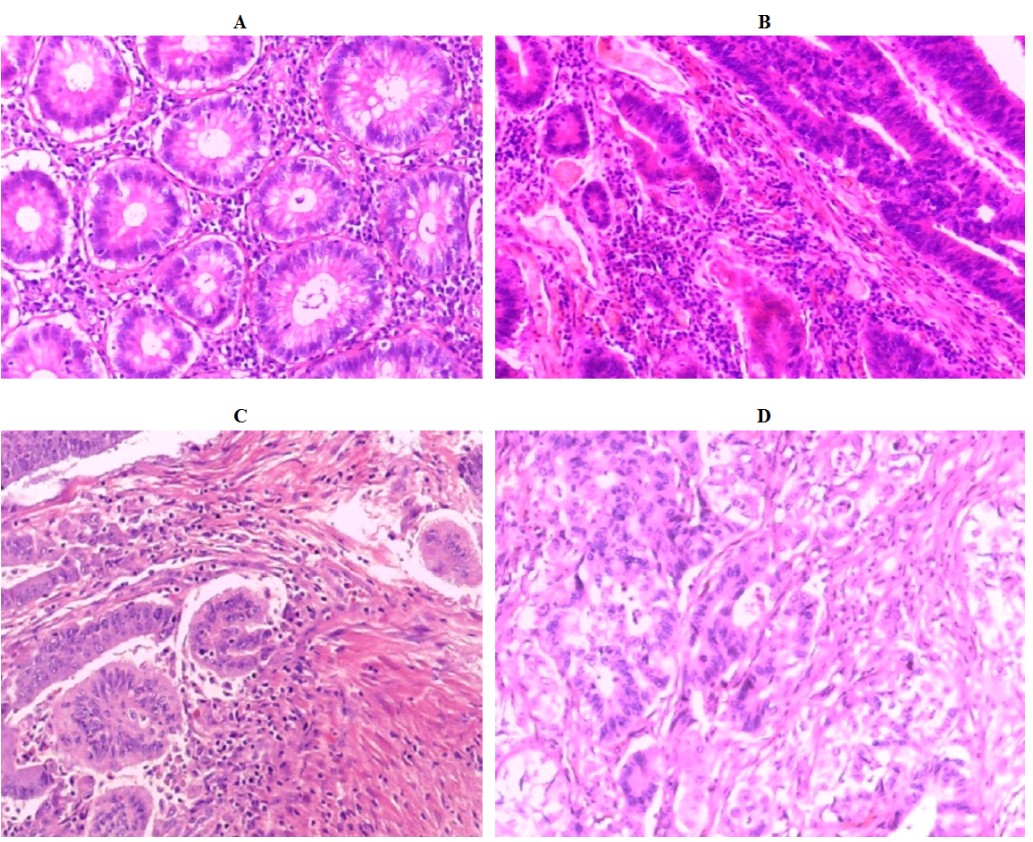

**Figure 1** **Colon histopathology images from the Stoean benchmark dataset (*Stoean et al., 2016*) with 40 × magnification factor: (A) normal (Grade 0), (B) cancer grade 1 (G1), (C) cancer grade 2 (G2), and (D) cancer grade 3 (G3).**

and the softmax represent the neural networks classifiers (*Alzubi, 2022*). CNN has the ability to extract the features from images by parameter tuning of the convolutional and the pooling layers. Thus, it achieves great success in many fields especially in medical image classifications such as skin disease (*Harangi, 2018*), breast (*Deniz et al., 2018*) and colon cancer classification (*Ghosh et al., 2021*). CNN is categorized into two approaches: either training from scratch or pre-trained models (*e.g.*, DenseNet (*Huang et al., 2017*), MobileNet (*Sandler et al., 2018*), and InceptionV3 (*Szegedy et al., 2016*). The most effective approach in medical image classification is the pretrained models due to the limited number of training samples (*Saini & Susan, 2020*).

CNN has been used in the domain of colon histopathlogical image classification. For example, *Postavaru et al. (2017)* utilized a CNN approach for the automated diagnosis of a set of colorectal cancer histopathological slides. They utilized CNN with five convolutional layers and reported accuracy of 91.4%. *Stoean (2020)* extended the work (*Postavaru et al., 2017*) and presented a modality method to tune the convolutional of the deep CNN. She introduced two Evolutionary algorithms for CNN parametrization. She conducted the experiments on colorectal cancer (*Stoean et al., 2016*) and reported the highest accuracy of

92%. It was obtained from these studies that the CNN models exceeded the handcrafted features.

While the CNN achieves high performance especially on large dataset size, it struggles to make such performance on small dataset size (*Deniz et al., 2018*; *Mahbod et al., 2020*), and simply results in overfitting issue (*Zhao, Huang & Zhong, 2017*). To overcome this issue, the concept of transfer learning technique of pretrained CNN models is exploited for the classification of colon histopathlogical images. In practice, the transfer learning technique of the pretrained models exports knowledge from previously CNN that has been trained on the large dataset to the new task with small dataset (target dataset). There are two approaches to transfer learning of pretrained models in medical image classification: feature extraction and fine-tuning (*Benhammou et al., 2020*). The former method extracts features from any convolutional or pooling layers and removes the last FCC and softmax layers. While in the latter, the pretrained CNN models are adjusted for specific tasks. It is important to remember that the number of neurons in the final FC layer corresponds to the number of classes in the target dataset (*i.e.,* the number of colon types). Following this replacement, the whole pre-trained model is retrained (*Mahbod et al., 2020*; *Benhammou et al., 2020*; *Zhi et al., 2017*) or the last FC layers are retrained (*Benhammou et al., 2020*). Various pretrained models (*e.g.*, DenseNet (*Huang et al., 2017*), MobileNet (*Sandler et al., 2018*), VGG16 (*Simonyan & Zisserman, 2014*), and InceptionV3 (*Szegedy et al., 2016*) have been introduced in recent years. Each pretrained model is constructed based on several convolution layers and filter sizes to extract specific features from the input image. However, transferring the begotten experience from the source (ImageNet) to our target (colon images) led to the loss of some powerful features of histopathological image analysis (*Boumaraf et al., 2021*). For example, CNN pretrained AlexNet and GoogleNet models were used on the colon histopathological images classification (*Popa, 2021*). However, they achieved poor standard deviation results. However, using these pretrained models on the colon dataset needs a specific fine-tuning approach to achieve acceptable results.

To accommodate the pretrained CNN models to the colon image classification, we design a new set of transfer learning models (DenseNet (*Huang et al., 2017*), MobileNet (*Sandler et al., 2018*), VGG16 (*Simonyan & Zisserman, 2014*), and InceptionV3 (*Szegedy et al., 2016*) to refine the pretrained models on the colon histopathological image tasks. Our transfer learning methods are based on a block-wise fine-tuning policy. We make the last set of residual blocks of the deep network models more domain-specific to our target colon dataset by adding dense layers and dropout layers while freezing the remaining initial blocks in the deep pretrained model. The adaptability of the proposed method is further extended by fine-tuning the neural network's hyper-parameters to improve the model generalization ability. Besides, a single pretrained model has a limited capacity to extract complete discriminating features, resulting in an inadequate representation of the colon histopathology performance (*Yang et al., 2019*). As a result, this study proposes an ensemble of pretrained CNN models architectures (E-CNN) to identify the representation of colon pathological images from various viewpoints for more effective classification tasks.

In this research, the following contributions are made:

- Investigate the influence of the standard TL approaches (DenseNet, MobileNet, VGG16, and InceptionV3) on the colon cancer classification task.
- Design a new set of transfer learning methods based on a block-wise fine-tuning approach to learn the powerful features of the colon histopathology images. The new design includes adding a set of dense and dropout layers while freezing the remainder of the initial layers in the pretrained models (DenseNet, MobileNet, VGG16, and InceptionV3) to make them more specific for the colon domain requirements.
- Define and optimize a set of hyper-parameters for the new set of pretrained CNN models to classify colon histopathological images.
- An ensemble (E-CNN) was proposed to extract complementary features in colon histopathology images by using an ensemble of all the introduced transfer learning methods (base classifiers). The proposed E-CNN merges the decisions of all base classifiers *via* majority voting and product rules.

The remainder of this research is organized as follows. The Literature Review section goes over the related works. Our proposed methodology is presented in detail in the Methodology section. The experiments Results and Discussion section analyzes and discusses the experimental results. The Conclusion brings this study to a close by outlining some research trends and viewpoints.

## LITERATURE REVIEW

Deep learning pretrained models have made incredible progress in various kinds of medical image processing, specifically histopathological images, as they can automatically extract abstract and complex features from the input images (*Manna et al., 2021*). Recently, CNN models based on deep learning design are dominant techniques in the CADs of cancer histopathological image classification (*Kumar et al., 2020*; *Mahbod et al., 2020*; *Albashish et al., 2021*). CNN learn high- and mid-level abstraction, which is obtained from input RGB images. Thus, developing CADs using deep learning and image processing routines can assist pathologists in classifying colon cancer histopathological images with better diagnostic performance and less computational time. Numerous CADs for identifying colorectal cancer using histological images had been introduced by a number of researchers in past years. These CADs vary from conventional machine learning algorithms of the deep CNN. In this study, we present the related work of the colorectal cancer classification relying on colorectal cancer dataset (*Stoean et al., 2016*) as real-world test cases.

The authors in *Postavaru et al. (2017)* designed a CNN model for colon cancer classification based on colorectal histopathological slides belonging to a healthy case and three different cancer grades (1, 2, and 3). They used an input image with the size of $256 \times 256 \times 3$. They created five convolutional neural networks, followed by the ReLU activation function. In the introduced CNN, various kernel sizes were utilized in each convolutional layer. Besides, they utilized batch normalization and only two FCC layers. They reported 91% accuracy in multiclass classification for the colon dataset in *Stoean et al. (2016)*. However, in the proposed approach, only the size of the kernels is considered, while other parameters, like learning rate and epoch size, were not taken into account.

The author in *Stoean (2020)* extended the previous study (*Postavaru et al., 2017*) by applying an evolutionary algorithm (EA) in the CNN architecture. This was to automate two tasks: first, EA was conducted for tuning the CNN hyper-parameters of the convolutional layers. Stoean determined the number of kernels in CNN and their size. Second, the EA was used to support SVM in parameters ranking to determine the variable importance within the hyper-parameterization of CNN. The proposed approach achieved 92% colorectal cancer grading accuracy on the dataset in *Stoean et al. (2016)*. However, using EA does not guarantee any diversity among the obtained hyper-parameters (solutions) (*Bhargava, 2013*). Thus, choosing the kernel size and depth of CNN may not ensure high accuracy.

In another study for colon classification but on a different benchmark dataset, the authors in *Malik et al. (2019)* have proved that the transferred learning from a pre-trained deep CNN model using InceptionV3 on a colon dataset with fine-tuning provides efficient results. Their methodology was mainly constructed based on InceptionV3. Then, the authors modified the last FCC layers to become harmonious with the number of the classes in the colon classification task. Moreover, the adaptive CNN implementation was proposed to improve the performance of CNN architecture for the colon cancer detection task. The study achieved around 87% accuracy for the multiclass classification task.

In another study (*Dif & Elberrichi, 2020a*), a framework was proposed for the colon histopathological image classification task. The authors employed a CNN based on transferred learning from Resnet121 generating a set of models followed by a dynamic model selection using the particle swarm optimization (PSO) metaheuristic. The selected models were then combined by a majority vote and achieved 94.52% accuracy on the colon histopathological dataset (*Kather et al., 2016*). In the same context, the authors in *Dif & Elberrichi (2020b)* explored the efficiency of reusing pre-trained models on histopathological images dataset instead of ImageNet based models for transfer learning. For this target, a fine-tuning method was presented to share the knowledge among different histopathological CNN models. The basic model was created by training InceptionV3 from scratch on one dataset while transfer learning and fine-tuning were performed using another dataset. However, this transfer learning-based strategy offered poor results on the colon histopathological images due to the limited number of the training dataset.

The conventional machine learning techniques have been utilized for the colon histopathology images dataset to achieve accepted results. For example, the 4-class colon cancer classification task on the dataset in *Stoean et al. (2016)* was utilized in *Boruz & Stoean (2018)* and *Khadilkar (2021)* to discriminate between various cancer types. In the former case (*Boruz & Stoean, 2018*), the authors extracted contour low-level image features from grayscale transformed images. Then, these features were used to train the SVM classifier. Despite its simplicity, the study displayed a comparable performance to some computationally expensive approaches. The authors reported accuracy averages between 84.1% and 92.6% for the different classes. However, transforming the input images to grayscale leads to losing some information and degrades the classification results. However, using thresholding needs fine-tuning, which is a complex task. In latter case (*Khadilkar, 2021*), the authors extracted morphological features from the colon dataset. Mainly, they extracted Harris corner and Gabor wavelet features. These features were

then used to feed the neural network classifier. The authors utilized their framework to discriminate between benign and malignant cases. However, they ignored the multiclass classification task, which is more complex task in this domain.

Most of the above studies utilized a single deep CNN (aka weak learner) model to address various colon histopathology images classification tasks (binary or multiclass). Despite their extensive use, a single CNN model has the restricted power to capture discriminative features from colon histopathology images, resulting in unsatisfactory classification accuracy (*Yang et al., 2019*). Thus, merging a group of weak learners forms an ensemble learning model, which is likely to be a strong learner and moderate the shortcomings of the weak learners (*Qasem et al., 2022*).

Ensemble learning of deep pretrained models has been designed to fuse the decisions of different weak learners (individuals) to increase classification performance (*Xue et al., 2020*; *Zhou et al., 2021*). A limited number of studies applied ensemble learning with deep CNN models on colon histopathological image classification tasks (*Popa, 2021*; *Lichtblau & Stoean, 2019*; *Rachapudi & Lavanya Devi, 2021*).

The authors in *Popa (2021)* proposed a new framework for the colon multiclass classification task. They employed CNN pretrained AlexNet and GoogleNet models followed by softmax activation layers to handle the 4-class classification task. The best-reported accuracies on *Stoean et al. (2016)* dataset ranged between 85% and 89%. However, the standard deviation of these results was around 4%. This means the results were not stable. AlexNet was also used in *Lichtblau & Stoean (2019)* as a feature extractor for the colon dataset. Then, an ensemble of five classifiers was built. The obtained results for this ensemble achieved around 87% accuracy.

In *Ohata et al. (2021)*, the authors use CNN to extract features of colorectal histological images. They employed various pretrained models, *i.e.,* VGG16 and Inception, to extract deep features from the input images. Then, they employed ensemble learning by utilizing five classifiers (SVM, Bayes, KNN, MLP, and Random Forest) to classify the input images. They reported 92.083% accuracy on the colon histological images dataset in *Kather et al. (2016)*. A research study in *Rachapudi & Lavanya Devi (2021)* proposed light weighted CNN architecture. RGB-colored images of colorectal cancer histology dataset (*Kather et al., 2016*) belonging to eight different classes were used to train this CNN model. It consists of 16 convolutional layers, five dropout layers, five max-pooling layers, and one FCC layer. This architecture exhibited high performance in term of incorrect classification compared to existing CNN models. Using ensemble learning model achieved around 77% accuracy (error of 22%).

Overall, the earlier studies, summarized in Table 1, revealed a notable trend in using deep CNN to classify colon cancer histopathological images. It was used to provide much higher performance than the conventional machine learning models. Nevertheless, training CNN models are not that trivial as they need considerable memory resources and computation and are usually hampered by over-fitting problems. Besides, they require a large amount of training dataset. In this regard, the recent studies (*Ahmad, Farooq & Ghani, 2021*; *Boumaraf et al., 2021*) have demonstrated that sufficient fine-tuned pretrained CNN models performance is much more reliable than the one trained from scratch, or in the

**Table 1  Summary of the major classification studies on colon cancer.**

| Authors in | Dataset used | CNN architecture | Accuracy | Using pretrained either feature extraction/fine- tuning |
|---|---|---|---|---|
| *Stoean (2020)* | colorectal in *Stoean et al. (2016)* | CNN model from scratch | 92% | Fine-tune: only kernel size and number of kernels in CNN using EA method |
| *Popa (2021)* | colorectal in *Stoean et al. (2016)* | AlexNet and GoogleNet | 89% | feature extractor |
| *Postavaru et al. (2017)* | colorectal in *Stoean et al. (2016)* | CNN model from scratch | 91% | The number of filters and the kernel size |
| *Lichtblau & Stoean (2019)* | colorectal in *Stoean et al. (2016)* | AlexNet | 87% | Feature extractor with ensemble learning |
| *Ohata et al. (2021)* | colorectal in *Kather et al. (2016)* | Set of pretrained models (VGG16, Inception, Resent) | 92.083% | Feature extraction |
| *Rachapudi & Lavanya Devi (2021)* | colorectal in *Kather et al. (2016)* | CNN architecture | 77% | Fine-tune CNN model |
| *Dif & Elberrichi (2020a)* | colorectal in *Kather et al. (2016)* | Pretrained Resnet121 | 94% | Feature extraction |
| *Boruz & Stoean (2018)* | colorectal in *Stoean et al. (2016)* | Contour low-level image features | 92.6% | |

worst cases the same. Besides, using ensemble learning of pretrained models show effective results in various applications of image classification tasks. Therefore, this research fills the gap in the previous studies for colon histopathological images classification by introducing a set of transfer learning models based on Dense. Then, reap the benefits of the ensemble learning to fuse their decision.

# METHODOLOGY

This study constructs an ensemble of the pretrained models with fine-tuning for the colon diagnosis based on histopathological images. Mainly, four pretrained models (DenseNet121 MobileNetV2, InceptionV3, and VGG16) are fine-tuned, and then their predicted probabilities are fused to produce a final decision for a test/image. The pretrained models utilize transfer learning to mitigate these models' weights to handle a similar classification task. Ensemble learning of pretrained models attains superior performance for histopathological image classification.

## Transfer learning (TL) and pretrained deep learning models for medical image

Transferring knowledge from one expert to another is known as transfer learning. In deep learning techniques, this approach is utilized where the CNN is trained on the base dataset (source domain), which has a large number of samples (*e.g.*, ImageNet). Then, the weights of the convolutional layers are transferred to the new small dataset (target domain). Using pretrained models for classification tasks can be divided into two main scenarios: freezing the layers of the pretrained model and fine-tuning the models. In the former scenario: the convolutional layers of a deep CNN model are frozen, and the last FCC are omitted. In this

way, the convolutional layers act as feature extractions. Then these features are passed to a specific classifier (*e.g.*, KNN, SVM) (*Taspinar, Cinar & Koklu, 2021*). While in the latter case, the layers are fine-tuned, and some hyper-parameters are adjusted to handle a new task. Besides, the top layer (fully connected layer) is adjusted for the target domain. In this study, for example, we configure the number of neurons in this layer four in accordance with the number of classes in the colon dataset. TL aims to boost the target field's accuracy (*i.e.,* colon histopathological) by taking full advantage of the source field (*i.e.,* ImageNet). Therefore, in this study, we transfer the weights of the set of four powerful pretrained CNN models (DenseNet (*Huang et al., 2017*), MobileNet (*Sandler et al., 2018*), VGG16 (*Simonyan & Zisserman, 2014*), and InceptionV3 (*Szegedy et al., 2016*)) with fine-tuning to increase the diagnosis performance of the colon histopathological image classification. The pretrained Deep CNN models and the proposed ensemble learning are presented in the subsequent section.

### Pretrained DenseNet121

Dense CNN(DenseNet) was offered by *Huang et al. (2017)*. The architecture of DenseNet was improved based on the ResNet model. The prominent architecture of DenseNet is based on connecting the model using dense connection instead of direct connection within all the hidden layers of the CNN (*Alzubaidi et al., 2021*). The crucial benefits of such an architecture are that the extracted features/features map is shared with the model. The number of training parameters is low compared with other CNN models similar to CNN models because of the direct synchronization of the features to all following layers. Thus, the DenseNet reutilizes the features and makes their structure more efficient. As a result, the performance of the DenseNet is increased (*Ahmad, Farooq & Ghani, 2021*; *Ghosh et al., 2021*). The main components of the DenseNet are: the primary composition layer, followed by the ReLU activation function, and dense blocks. The final layer is a set of FC layers (*Talo, 2019*).

### Pretrained MobileNetV2

MobileNet (*Sandler et al., 2018*) is a lightweight CNN model based on inverted residuals and a linear bottleneck, which form shortcut connections between the thin layers. It is designed to handle limited hardware resources because it is a low-latency model, and a small low power. The main advantage of the MobileNet is the tradeoff between various factors such as latency, accuracy, and resolution (*Krishnamurthy et al., 2021*). In MobileNet, depth separable convolutional (DSC) and point-wise convolutional kernels are used to produce feature maps. Predominantly, DSC is a factorization approach, which replaces the standard convolution with a faster one. In MobileNet, DSC first uses depth-wise kennels 2-D filters to filter the spatial dimensions of the input image. The size of the depth-wise filter is Dk $\times$Dk $\times$1, where Dk is the size of the filter, which is much less than the size of the input images. Then, it is followed by a point-wise convolutional filter that mainly applied to filter the depth dimension of the input images. The size of the depth filter is $1 \times 1 \times n$, where n is the number of kernels. They separate each DSC from point-wise convolutional using batch normalization and ReLU function. Therefore, DSC is called (separable). Finally, the last FCC is connected with the Softmax layer to produce the final output/ classification

result. Using depth-wise convolutional can reduce the complexity by around 22.7%. This means the DSC takes only approximately 22% of the computation required by the standard convolutional. Based on this reduction, MobileNet is becoming seven times faster than the traditional convolutional. Thus, it becomes more desirable when the hardware is limited (*Srinivasu et al., 2021*).

### Pretrained InceptionV3

Google teams in *Szegedy et al. (2016)* introduced the InceptionV3 CNN. The architecture of InceptionV3 was updated based on the inceptionV1 model, as illustrated in Fig. 2. It mainly addressed some issues in the previous inceptionV1 such as auxiliary classifiers by add batch normalization and representation bottleneck by adding kernel factorization (*Mishra et al., 2020*). The architecture of the InceptionV3 includes multiple various types of kernels (*i.e.,* kernel size) in the same level. This structure aims to solve the issue of extreme variation in the location of the salient regions in the input images under consideration (*Mishra et al., 2020*). The InceptionV3 (*Szegedy et al., 2016*) utilizes a small filter size ($1 \times 7$ and $1 \times 5$) rather than a large filter ($7 \times 7$ and $5 \times 5$). In addition, a bottleneck of $1 \times 1$ convolution is utilized. Therefore, better feature representation.

The architecture of InceptionV3 (*Szegedy et al., 2016*) is demonstrated in Fig. 2. It starts with input data (image), and then mapped parallel computations will be shaped into three different convolutional layers with $3 \times 3$ or $5 \times 5$ filter size. The output of these layers is aggregated into a single layer, which represents the output layer (*e.g.,* ensemble technique). Using parallel layers with each other will save a lot of memory and increase the model's capacity without increasing its depth.

### Pretrained VGG16

VGG16 was presented by *Simonyan & Zisserman (2014)* as a deeper convolutional neural network model. The basic design of this model is to replace the large kernels with smaller kernels, and extending the depth of the CNN model (*Alzubaidi et al., 2021*). Thus, the VGG16 becomes potentially more reliable in carrying out different classification tasks. Figure 3 shows the basic VGG16 (*Simonyan & Zisserman, 2014*) architecture. It consists of five blocks with 41 layers, where 16 layers have learnable weights; 13 convolutional layers and three FCC layers from the learnable layers (*Khan et al., 2020*). The first two blocks include two convolutional layers, while the last three blocks consist of three convolutional layers. The convolutional layers use small kernels with size of $3 \times 3$ and padding 1. These convolutional layers are separated using the max-pooling layers that use $2 \times 2$ filter size with padding 1. The output of the last convolutional layer is 4,096, which makes the number of neurons in the FCC 4,096 neurons. As illustrated in Table 2, VGG16 uses around 134 million parameters, which raises the complexity of VGG16 relating to other pretrained models (*Tripathi & Singh, 2020*; *Koklu, Cinar & Taspinar, 2022*).

## The proposed deep CNN ensemble based on softmax

The proposed deep ensemble CNNs (E-CNNs) architecture is based on two phases (base classifiers and fuse techniques). In the former phase, four modified models have been utilized: DenseNet121, MobileNetV2, InceptionV3, and VGG16 pretrained CNN classifier.

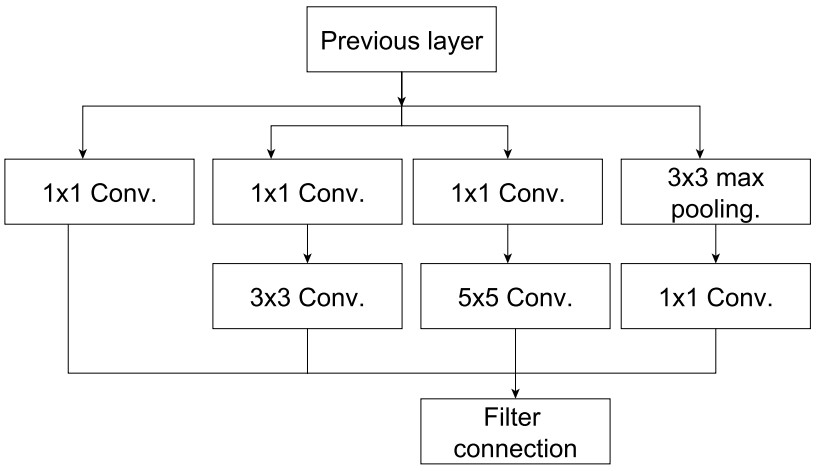

**Figure 2** The inception model from *Talo (2019)*.

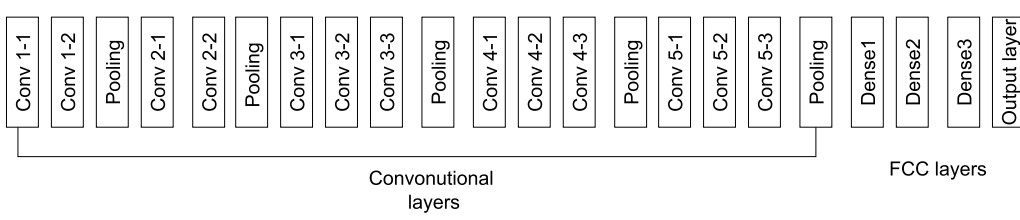

**Figure 3** The VGG16 model (*Simonyan & Zisserman, 2014*).

**Table 2** Summary of deep architectures used in this work.

| Architecture | No. of Conv layers | No. of FCC layers | No. of training parameters | Minimum image size | Number of extracted features | Top 5 error on ImageNet |
|---|---|---|---|---|---|---|
| DenseNet121 | 120 | 1 | 7 million | 221×221 | 1,024 | 7.71% |
| InceptionV3 | 42 | 1 | 22 million | 299 × 299 | 2,048 | 3.08% |
| VGG16 | 13 | 3 | 134 million | 227 × 227 | 4,096 | 7.30% |
| MobileNet | 53 | 3 | 3.4 million | 224 × 224 | 1,024 | -% |

While the latter phase focuses on combining the decisions of the base classifiers (in the first phase). Two types of fusion techniques have been employed in the proposed E-CNNS: majority voting and product rule. On the one hand, the majority voting is based on the prediction value of the base classifier. On the other hand, the product rule is based on the probabilities of the base classifiers (*i.e.,* pretrained model), the details of the proposed E-CNNs are as follows:

### The proposed modified deep pretrained models

After adapting the four pretrained models (DenseNet121, MobileNetV2, InceptionV3, and VGG16), they serve as base classifiers in the proposed E-CNN for the automated

classification of colon H&E histopathological images. The standard previous pretrained models extract various features from the training images to discriminate between different types of cancer (multiple classes) in the colon images dataset. However, each pretrained model is based on a set of convolution layers and filter sizes to extract different features from the input images. As a result, no pretrained model can be more general in extracting all the distinguishing features from the input training images (*Ghosh et al., 2021*).

However, using initial weights in pretrained models affect the classification performance because the CNNs pretrained models are nonlinear designs. These pretrained models learn complicated associations from training data with the assistance of back propagation and stochastic optimization (*Ahmad, Farooq & Ghani, 2021*). Therefore, this study introduces a block-wise fine-tuning technique to adapt the standard CNNs models to handle the heterogeneity nature in colorectal histology image classification tasks.

Figure 4 illustrates the main steps of the design of the block-wise fine-tuning technique. First, the benchmark colon images are loaded. Then, some preprocessing tasks on the training and testing images are performed to prepare them for the pretrained models, (*e.g.,* resizing them to $224 \times 224 \times 3$). The images are then rescaled to 1/255 as in the previous related studies (*Szegedy et al., 2016*). After splitting the dataset into training and testing, the four independent pretrained models: (*Huang et al., 2017*), MobileNet (*Sandler et al., 2018*), VGG16 (*Simonyan & Zisserman, 2014*), and InceptionV3 (*Szegedy et al., 2016*) are loaded without changing their weights. Then, the FCC and softmax layers are omitted from the loaded pretrained CNN models. These layers were originally designed to output 1,000 classes from the ImageNet dataset. Two dense layers with a varying number of hidden neurons are then added to strengthen the vital data-articular feature learning from each individual pretrained model. These dense layers are followed by the ReLU nonlinear activation function, which allows us to learn complex relationships among the data (*Ahmad, Farooq & Ghani, 2021*; *Garbin, Zhu & Marques, 2020*). Next, a 0.3 dropout layer is added to address the long training time and overfitting issues in classification tasks (*Deniz et al., 2018*; *Boumaraf et al., 2021*). At the end of each pretrained model, the last FCC with the softmax layer is added. The FCC is simply a feed-forward neural network, which is fed by flattened input from the last pooling layer of the pretrained model. In this study, based on the number of classes in this work, the number of neurons in FCC is set to four instead of 1000 classes of ImageNet. While the softmax layer (activation layer) is inserted on top of each model to train the obtained features and produce the classification output based on max probability. Algorithm 1 shows the main steps of the block-wise fine-tuning technique for each individual model in the proposed E-CNNs.

### Ensemble fusing methods for the proposed E-CNN

This study introduces E-CNNs based on the four modified CNN pretrained models (DenseNet (*Huang et al., 2017*), MobileNet (*Sandler et al., 2018*), VGG16 (*Simonyan & Zisserman, 2014*), and InceptionV3 (*Szegedy et al., 2016*)) for the automated classification of colon H&E histopathological images. The four adaptive models are trained on the training dataset. Then evaluated on the tested dataset. The output probabilities of the four pretrained models are connected to produce 16-D feature vector (*i.e.,* each individual with

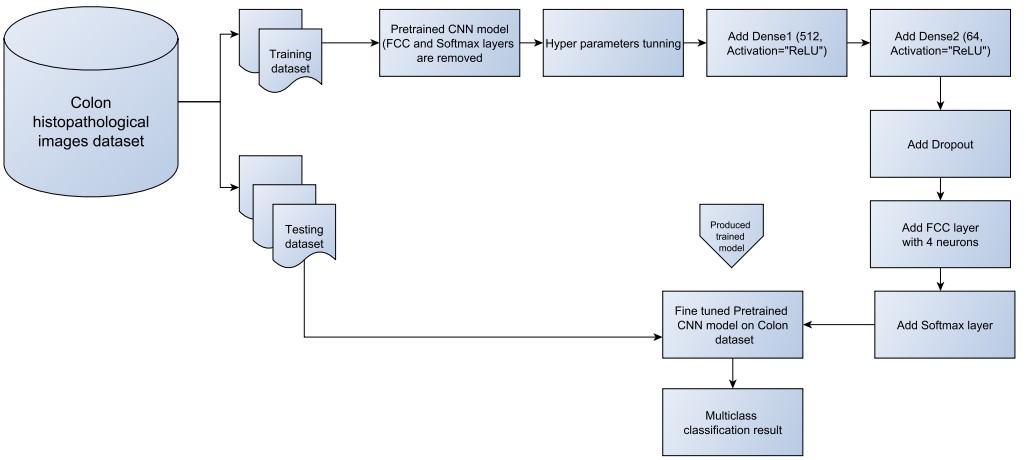

**Figure 4** Block diagram of the proposed block-wise fine-tuning for each pretrained model from (DenseNet121, MobileNetV2, InceptionV3, and VGG16).

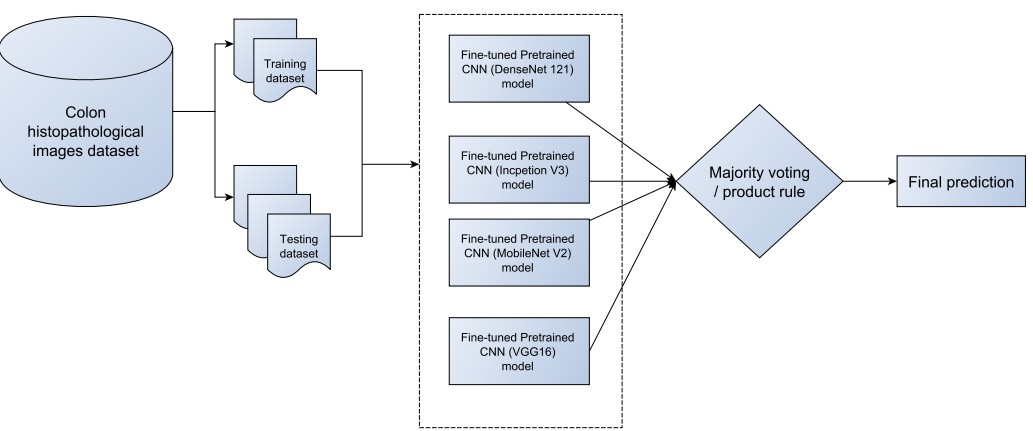

**Figure 5** The proposed E-CNNS with the four adaptive pretrained models (DenseNet121, MobileNetV2, InceptionV3, and VGG16).

its softmax produce four probabilities based on the number of classes in colon images). Then, various combination methods (majority voting, and product rule) are employed to produce a final decision for the test image. Figure 5 illustrates the proposed ECNNs with the merging techniques (ECNN (product rule) and E-CNN (majority voting)).

In the majority voting technique, each base classifier allocates a class label output (*i.e.,* a predicted label) to the provided test sample. It counts the votes of all the class labels from the base classifiers. Then, the class that obtains the maximum number of votes is nominated as the final decision for the E-CNN (majority voting) as described in Eq. (1).

$$P(I) = \max_{j=1 \, to \, c} \sum_{t=1}^{T=4} d_{tj} \tag{1}$$

where $P(I)$ denotes the final decision for the test image (I), $t$ denotes a base classifier, four base classifiers are utilized ($T = 4$), $c$ is the class label. $d_{tj}$ denotes the class label $j$ for the $I$ by a base classifier $t$. The final decision of the input test image $I$ is the class $j$ which has the maximum occurrence.

While in the product rule, the posterior probability outputs $P_t^j(I)$ for each class label j are generated by the base classifier $t$ for the test image (I). Then the class with the maximum likelihood of product is considered the final decision. Equation (2) shows the product rule technique in the proposed E-CNN (product rule). Algorithm 2 illustrates the proposed E-CNNs with majority voting and product rule.

$$P(I) = \max_{j=1 \, to \, c} \prod_{t=1}^{T=4} P_t^j(I) \tag{2}$$

Based on Algorithms 1, 2, and Figs. 4 and 5, the following points are taken into account: First, the CNN model is adapted to handle the heterogeneity in the colon histopathological images using the Block-wise fine-tuning technique for each of the pretrained models. It extracts additional abstract features from the image that aid in increasing intra-class discrimination. Second, ensemble learning is employed to improve the performance of the four adaptive pretrained models. As a result, the final decision regarding the test images will be more precise.

---

**Algorithm 1** Building and training the adaptive pretrained models [Block-wise fine-tuning for each pretrained model].

---

1: input:Training data(T), N samples: T = [$x_1, x_2, \ldots, x_N$], with Category: y = [$y_1, y_2, \ldots, y_N$], pretrained CNN models( M), M=[ DenseNet121, MobileNetV2, InceptionV3, and VGG16 models].

2: **for** each $I$ in $M$ **do**

3:     Remove the last FCC and softmax layers

4:     Add Dense1 layer with number of neurons equal to 512 and activation function=ReLU

5:     Add Dense2 layer with number of neurons equal to 64 and activation function=ReLU

6:     Add dropout layer

7:     Add FCC layers with number of neurons equal to 4( based on number classes in the colon dataset)

8:     Add softmax layer ( for output probabilities)

9:     Initialize the Hyper-parameters values, as listed in Table **??**

10:     Build the final model ($adaptI$)

11:     Train the $adapI$ on $T$

12:     Append $adapI$ into $adapM$

13: **end for**

14: Output: Adaptive models ($adaptM$), adaptM=[ adapt_DenseNet121, adapt_MobileNetV2, adapt_InceptionV3, and adapt_VGG16 models]

---

---

**Algorithm 2** Ensemble of adaptive models and evaluating the ensemble model on test colon histopathlogical images.

---

1: Input: Adaptive models ($adaptM$), $adaptM$=[ adapt_DenseNet121, adapt_MobileNetV2, adapt_InceptionV3, and adapt_VGG16 models], Test images set( D), with z samples: R = [$x_1, x_2, x_3, \ldots, x_z$], with Category: y = [$y_1, y_2, \ldots, y_z$]

2: **for** $j$ in $D$ **do**

3:     **for** each individual $I$ in $adaptM$ **do**

4:         Evaluate the performance of I using the test data $j$.

5:         $P[j, I]$=probabilities of each class for the test image $j$ when using the individual $I$.

6:         $V[j, I]$=prediction for the test image $j$ when using the individual $I$.

7:     **end for**

8:     Compute the ensemble final prediction for test image $j$ based on majority voting and $V[j, :]$ (ECNN(majority_voting))

9:     Compute the ensemble final prediction for test image $j$ based on the product rule(ECNN(product_rule)) and $p[j, :]$

10:     Output: class prediction for $D$ using (ECNN(majority_voting)) and (ECNN(product_rule))

11: **end for**

---

### Resources used

All the experiments are implemented using TensorFlow, Keras API, and utilized Python programming in Google Colaboratory or CoLab. In the CoLab, we utilize Tesla GPU to run our experiment after loading the dataset into the Google drive (*Postavaru et al., 2017*).

## EXPERIMENTS RESULTS AND DISCUSSION

This section outlines the experiments and evaluation results from the (E-CNN) and its individual models presented in this research. This section also entails a synopsis of the training and test datasets. The results using the proposed E-CNN, with majority voting and product rule, other standard pretrained models, and state-of-the-art colon cancer classification methods are also presented in this section. Comparisons between the proposed E-CNN and other CNN models from scratch are presented in this section.

### Dataset

To evaluate the validity of the proposed E-CNN for colon diagnosis from histopathological images, two distinct benchmarks colon histology images datasets from (*Stoean et al., 2016*; *Kather et al., 2016*) are applied. Further information about these datasets is as follows:

(A) Stoean (370 images): The histology images dataset (*Stoean et al., 2016*) were collected from the Hospital of Craiova, Romania. The benchmark dataset consist of 357 histopathological H&E of normal grade (grade 0) and for cancer grades (grades 1, 2, and 3), with 10× magnification. They have a similar 800 × 600 pixels resolution. The images' distribution for the classes is as follows: Grade 0: 62 images, grade 1: 96 images, grade 2: 99 images, and grade 3: 100 images. All images are RGB color 8-bit depth with

JPEG format. Figure 1 shows some samples from the images and how they are close to each other in the structure, which discriminates between various complicated grades.

(B) Kather (5,000 images): The dataset (*Kather et al., 2016*) includes 5,000 histology images of human colon cancer. The samples were gathered from the Institute of Pathology, University Medical Center, Mannheim, Germany. The benchmark dataset consists of histopathological H&E of eight classes: namely ADIPOSE, STROMA, TUMOR, DEBRIS, MUCOSA, COMPLEX, EMPTY, and LYMPHO. Each class consists of 625 images with a size of 150 × 150 pixels, 20 × magnification, and RGB channel format.

## Experimental setting

As the proposed E-CNN aims to assist in diagnosing colon cancer based on the histopathological images, the benchmark dataset in *Stoean et al. (2016)* is considered during the experiments' work. The dataset was divided into 80% training and 20% testing. In E-CNN, the Hyperparameters, as illustrated in Table 3, were fine-tuned with the same setting for all the proposed transfer learning models. The training and testing images were resized to 224 × 224 for comfort with the proposed transform learning models. The batch size was chosen as 16; the minimum learning rate was specified as min_lr = 0.000001. The learning rate was determined to be small enough to slow down learning in the models (*Popa, 2021*; *Kaur & Gandhi, 2020*). The number of epochs was selected as 10. These models were trained by stochastic gradient descent (SGD) with momentum. All the proposed TL models emploed the cross entrotpy (CE) as the loss function. The cross-entropy is mainly utilized to estimate the distance between the prediction likelihood vector(E) and the one-hot-encoded ground truth label(T) (*Boumaraf et al., 2021*) probability vector. The following equation depicts the CE Eq. (3):

$$CE(E, T) = -\sum_{t=1} T_i \log E_i \tag{3}$$

where *CE* is used to tell how well the output E matches the ground truth T. Furthermore, the dropout layer was added to all the proposed TL models to avoid over-fitting affair during training. As a result, it drops the activation randomly during the training phase and avoiding units from over co-adapting (*Boumaraf et al., 2021*). In this study, dropout was set to 0.3 to randomly drop out the units with a probability of 0.3, which is typical when introducing the dropout in deep learning models.

## Evaluation criteria

In this work, multiclass (four-class) classification tasks have been carried out using the base classifiers and their ensembles on the benchmark colon dataset (*Stoean et al., 2016*). The obtained results have been evaluated using average accuracy, average sensitivity, average specificity, and standard deviation over ten runs. All of these metrics are counted based on the confusion matrix, which includes the true negative (TN) and true positive (TP) values. TN and TP symbolize the acceptably classified benign and malignant samples, respectively. The false negative(FN), and false positive (FP) denote the wrong classified malignant and benign samples. These metrics are designed as follows:

**Table 3 Hyperparameters used in the proposed individual transfer learning models and an ensemble model.**

| Hyperparameters | Value |
| --- | --- |
| Image size | $224 \times 224$ |
| Optimizer | 0.005 |
| Maximum Habitat probability | SGD with momentum |
| Learning rate | 1e−6 |
| Batch size | 16 |
| Number of epochs | 10 |
| Dropout | 0.3 |
| Loss function | Cross Entropy |

- The average classification accuracy: The correctly categorized TP and TN numbers combined with the criterion parameter, are generally referred to as accuracy. A technique's classification accuracy is measured in Eq. (4) as follows:

$$Acc = \frac{1}{M} \sum_{j=1}^{M} \frac{TP + TN}{TP + TN + FP + FN} * 100\%, \tag{4}$$

where $M$ is the number of independent runs of the proposed ECNN with its individual.
- Average sensitivity: Sensitivity is also called recall. It represents the proportion of positive samples, which are efficiently determined as described in Eq. (5):

$$Sensitivity = \frac{1}{M} \sum_{j=1}^{M} \frac{TP}{TP + FN} * 100\%, \tag{5}$$

The sensitivity value is between [0, 1] scale. One shows the ideal classification, while zero shows the worst classification possible. Multiplication by 100 is applied on the sensitivity to obtain the required percentage.
- Average Specificity: Specificity represents an evaluation metric that is provided for negative samples within a classification approach. In particular, it attempts to measure the negative samples' proportion, which is efficiently classified. Specificity is computed as Eq. (6):

$$Specificity = \frac{1}{M} \sum_{j=1}^{M} \frac{TN}{TN + FP} * 100\% \tag{6}$$

## Results and discussion

This subsection presents the experimental results obtained from the proposed E-CNN and its individuals. These results are compared to the classification accuracy results using standard pretrained models (*e.g.*, DenseNet, MobileNet, VGG16, and InceptionV3). After that, the performance of the standard pretrained models was compared to the adaptive pretrained models' performance to evaluate the influence of block-wise fine-tuning policy. The proposed E-CNN was also compared with the state-of-the-art CNN models for colon

**Table 4** Evaluation results for the proposed E-CNN, its individuals (modified TL models) when number of epochs = 10, and the standard TL models on colon histopathlogical images dataset based on the average accuracy, sensitivity, specificity, and average standard deviation (STD) in 10 runs, best results in bold.

| Pretrained models | Accuracy | Sensitivity | Specificity |
|---|---|---|---|
| Standard DenseNet121 | 90.41 ± 3.1 | 91.25 ± 2.9 | 100 ± 0 |
| Standard MobileNetV2 | 90.27 ± 2.9 | 88.25 ± 1.9 | 99.23 ± 2.3 |
| Standard InceptionV3 | 87.12 ± 2.0 | 92.75 ± 2.0 | 100 ± 0 |
| Standard VGG16 | 62.19 ± 7.0 | 63.21 ± 7.3 | 100 ± 9.9 |
| Modified DenseNet121 | 92.32 ± 2.8 | 92.99 ± 2.8 | 100 ± 0.0 |
| Modified MobileNetV2 | 92.19 ± 3.8 | 90.75 ± 2.0 | 100 ± 0.0 |
| Modified InceptionV3 | 89.86 ± 2.2 | 95.0 ± 1.5 | 100 ± 0.0 |
| Modified VGG16 | 72.73 ± 3.9 | 73.0 ± 3.6 | 87.43 ± 12.4 |
| **Proposed E-CNN (product)** | **95.20 ± 1.64** | **95.62 ± 1.50** | **100 ± 0.0** |
| Proposed E-CNN (Majority voting) | 94.52 ± 1.73 | 95.0 ± 1.58 | 100 ± 0.0 |

cancer classification such as *Postavaru et al. (2017)*; *Stoean (2020)* and *Popa (2021)*. In the end, to assess the significance of the proposed E-CNN, statistical test methods were used to verify whether there is a statistically significant difference between the performance of the E-CNN and the performance of the state-of-the-art CNN models.

The experimental results of this study were built based on the average runs. Mainly, ten separate experiments were used to obtain the average value.The experiments were carried out on two benchmark colon histopathological image datasets: Stoean and Kather datasets, to test the robustness of the proposed methods. The former dataset is the Stoean dataset, which includes four different classes, mainly: benign, grade1, grade2, and grade3. While the second dataset (the Kather dataset) includes eight diverse classes, Each dataset was divided into 80% for the training set and 20% for the testing set. The results of classification performance in this study are for the test dataset. The The classification tasks were accomplished using individual classifiers of the modified transfer learning set (Modified DenseNet121, Modified MobileNetV2, Modified InceptionV3, and Modified VGG16). The softmax of the FCC of these transfer learning set is used as the classification algorithm. Then, the ensemble (E-CNN) was obtained *via* product and majority voting aggregation methods. To illustrate the proposed E-CNN performance, the average accuracy, sensitivity, and specificity over the ten runs are used for evaluating the testing dataset. Besides, the standard deviation (STD) for each base classifier and the E-CNN are also used to estimate the effectiveness of the proposed E-CNN. The experimental results of the proposed E-CNN and its individuals (*i.e.,* base classifiers) on the first dataset are shown in Tables 4, 5 and 6, and Figs. 6, 7, 8, 9, 10, and 11, respectively. Meanwhile, the results of the (Kather's) dataset (*Kather et al., 2016*) are shown in Table 7.

Table 4 compares the results obtained by the modified pretrained models, the baseline (standard) pretrained model, and the proposed (E-CNN). Table 4, indicates that the results obtained from all the modified models successfully outperformed the standard pretrained models on the first dataset. The highest classification success belongs to the modified DenseNet121 model. It achieved approximately 92.3% test accuracy, which was 2.0% more

**Table 5  Summary of the major classification studies on colon cancer.**

| Authors in | Dataset used | CNN architecture | Accuracy | T-test/*p*-value |
|---|---|---|---|---|
| *Stoean (2020)* | colorectal in *Stoean (2020)* | CNN model from scratch | 92% | *P* < 0.0001 |
| *Popa (2021)* | colorectal in *Stoean et al. (2016)* | AlexNet | 89.53% | *P* < 0.0001 |
| *Stoean et al. (2016)* | colorectal in *Stoean et al. (2016)* | GoogleNet | 85.62% | *P* < 0.0001 |
| *Postavaru et al. (2017)* | colorectal in *Stoean et al. (2016)* | CNN model from scratch | 91% | *P* < 0.0001 |
| **Proposed E-CNN (product rule)** | colorectal in *Stoean et al. (2016)* | **Modified TL models with ensemble learning (using product rule)** | **95.20%** | |
| Proposed E-CNN (Majority voting) | colorectal in *Stoean et al. (2016)* | Modified TL models with ensemble learning (using majority voting) | 94.52% | |

Notes.
Best results are shown in bold.

**Table 6  Valuation results for the proposed E-CNN, its individuals (modified TL models) when number of epochs = 30, and the standard TL models on colon histopathlogical images dataset based on the average accuracy, sensitivity, specificity, and average standard deviation (STD) in 10 runs, best results in bold.**

| Pretrained models | Accuracy | Sensitivity | Specificity |
|---|---|---|---|
| Modified DenseNet121 | 96.8. ± 2.7 | 97.0 ± 2.4 | 100 ± 0.0 |
| Modified MobileNetV2 | 94.48 ± 2.6 | 95.5 ± 1.9 | 100 ± 0.0 |
| Modified InceptionV3 | 94.52 ± 1.7 | 95.1 ± 1.2 | 100 ± 0.0 |
| Modified VGG16 | 79.4 ± 1.9 | 79.9 ± 3.6 | 100 ± 0.0 |
| **Proposed E-CNN (product)** | **97.2 ± 1.27** | **97.5 ± 1.8** | **100 ± 0.0** |
| Proposed E-CNN (Majority voting) | 95.89 ± 1.3 | 96.2 ± 1.57 | 100 ± 0.0 |

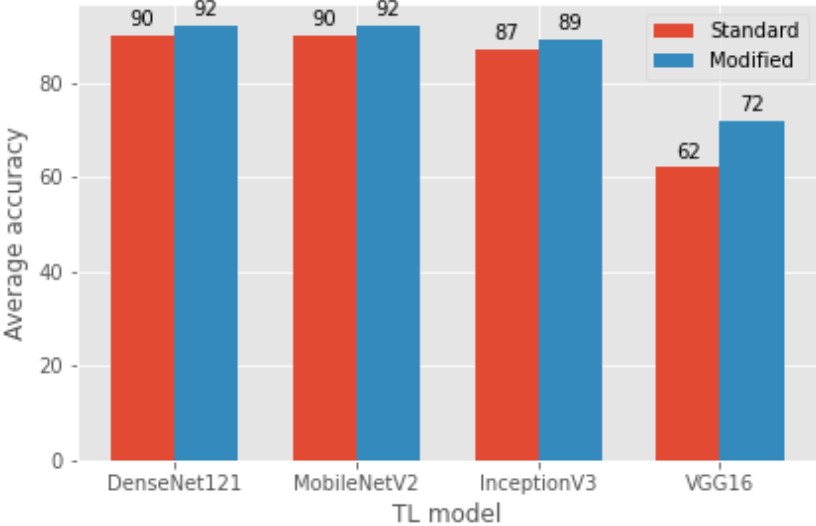

**Figure 6  A comparison of modified TL models with standard TL models (original) in terms of average classification accuracy.**

accurate than the standard DenseNet121. It is clear that the modified DenseNet121 model

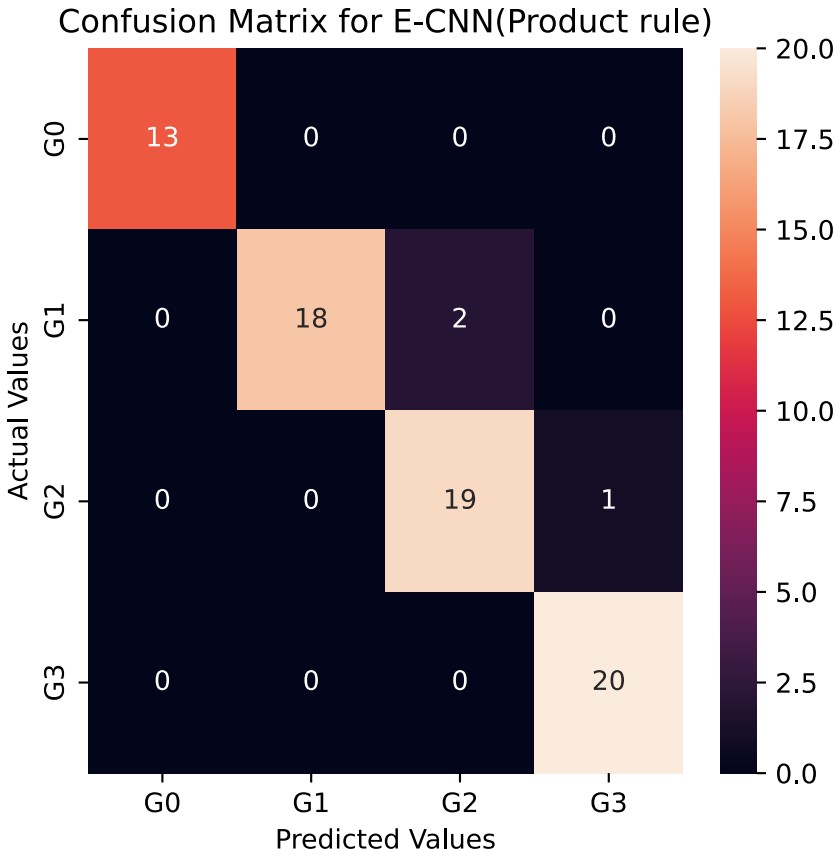

**Figure 7** Confusion matrix of the E-CNN (product rule) on the Stoean testing dataset when number of epochs = 10.

has the highest specificity and sensitivity metrics as in the classification success. This is due to the fine-tuned modified DenseNet121 architecture's custom design, which aids in extracting discriminating features from the input colon histopathological images and can distinguish between different classes in this domain. The second highest accuracy among the four modified pretrained models is the modified MobileNetV2. It achieved 92.19% test accuracy, which was comparable to the improved DenseNet121. In more details, the average accuracy rate difference between the modified MobileNetV2 and the standard MobileNetV2 is more than 2%.

However, among the four proposed individual pretrained models, the modified VGG16 is the least accurate, it rated about 79% for the multiclass classification task. This could be the explanation for VGG16's limited number of layers (*i.e.,* 16 layers). Compared to the standard VGG16, the average accuracy rate difference between the modified VGG16 and the standard VGG16 was more than 10%, which was big and statistically significant. This astounding level of performance of the modified models could be attributed to the ability of the adaptation layers to find the most abstract features, which aid the FCC and softmax classifier in discriminating between various grades in colon histopathological

images. As a result, it reduces the problem of inter-class classification. Moreover, the proposed modified pretrained models outperformed the standard models, boosting the decisions of these models and enabling them to achieve a better generalization ability than a single pretrained model (*Cao et al., 2020*). In this study, two ensemble learning models are utilized: E-CNN (product rule) and E-CNN (majority voting), to merge the decisions of the single models. The former is based on merging the probabilities of the individual modified models. While the latter is based on combining the output predictions of the individual, Fig. 7 confirms the confusion matrix obtained as a result of the tested samples (20% of the dataset) for one run of the classification performed through the proposed E-CNN (product rule). The empirical results of the proposed ECNN (majority voting) and E-CNN (product rule) achieved accuracy rates of 94.5% and 95.2%, respectively.

These accuracy values were higher compared to the individual models. For example, the E-CNN (product rule) result showed 3.2% increase compared to the modified DenseNet121 model. This result reveals the significance of the product rule in the proposed E-CNN for colon image classification because it is based on an independent event (*Albashish et al., 2016*). To show the adequacy of the proposed E-CNN, sensitivity was computed. Table 4 confirms that the E-CNN has higher sensitivity values than all the individual models. It is worth noting that the sensitivity performance level matches the accuracy values, thereby emphasizing the consistency of the E-CNN results. E-CNN (product rule) was able to yield a better sensitivity value (95.6%). Among all the proposed transfer learning models, InceptionV3 delivered the overall maximum sensitivity performance. Besides, the specificity measure shows that the E-CNN and its individuals are able to detect negative samples which are correctly classified for each class.

Furthermore, the standard deviation analysis over the ten runs shows that the ensemble E-CNNs (product rule) has the minimum value (around 1.7%). These results indicate that it is stable and capable of producing optimal outcomes regardless of the randomization.

To show the adequacy of the proposed modified CNN models even after being trained on a smaller dataset, we have provided accuracy and loss (error function) curves. The loss function quantifies the cost of a particular set of network parameters based on how often they generate output in comparison to the ground truth labels in the training set. The TL models employ SGD to determine the optimal set of parameters for minimizing the loss error. Figures 8 and 9 depict the proposed TL models' accuracies and loss curves for the training and testing datasets over ten epochs. Figure 8 shows that the proposed DenseNet and Inception models achieved good accuracy for the training and test datasets over various epochs, while the MobileNet and VGG15 models performed adequately. One possible explanation is that the proposed models are stable during the training phase, allowing them to converge to the best effect. The DenseNet121 loss curve indicates that the training loss dropped significantly much faster than the VGG16 and that the testing accuracy improved much faster. In more detail, the VGG16 loss function was linearly reduced, whereas the DenseNet loss function was dramatically reduced. This is consistent with DenseNet121's classification performance in Table 4, where it outperformed all other proposed models. Furthermore, one can see that all of the proposed TL models, except

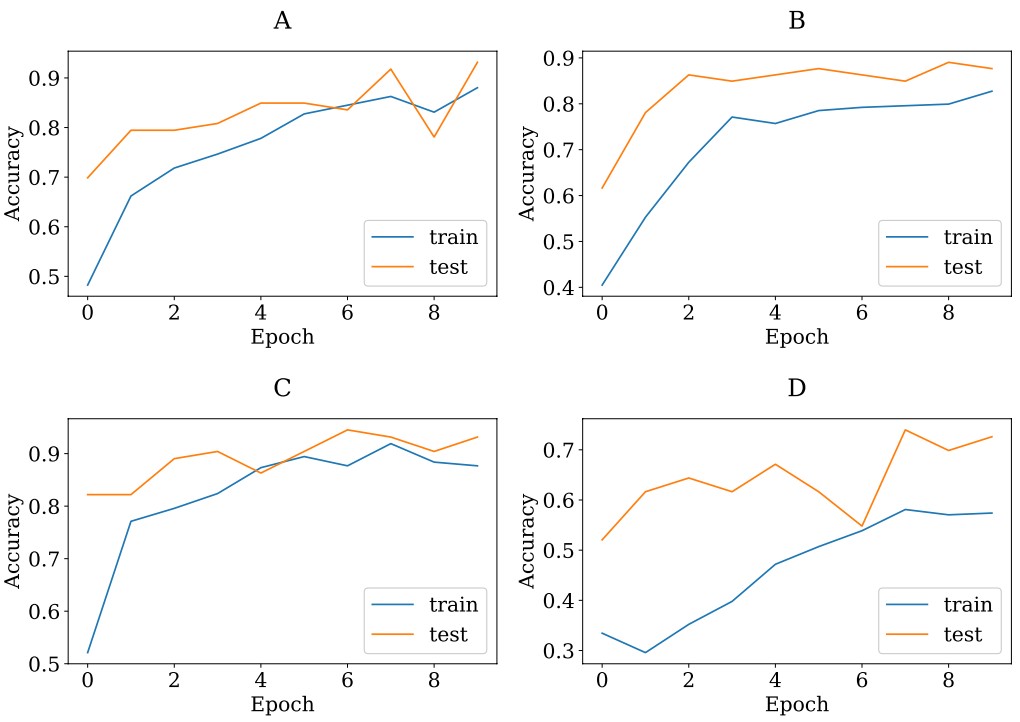

**Figure 8** The accuracy learning curves of training and testing derived from the four modified CNN base learners, when the number of epochs is ten on the colon histopathological image benchmark Stoean's dataset used in this study: (A) modified DenseNet121, (B) modified InceptionV3, (C) modified MobileNetV2, and (D) modified VGG16.

the VGG16, achieved high testing accuracies. These models improve the generalization performance simultaneously.

To further demonstrate the efficacy of the proposed E-CNNs, we also compare the obtained results on the colon histopathlogical images benchmark dataset with the most recent related works (*Stoean, 2020*; *Popa, 2021*). Table 5 contains the comparison between the proposed E-CNNs and the recent state-of-the-art studies. From the tabular results, one see that the proposed E-CNNs achieved higher results comparing either to pretrained models, such as in *Stoean et al. (2016)*, or to constructing CNN from scratch. One of the main reasons for these superior results is the adaptation of the transfer learning models with the appropriate layers; additionally, using ensemble learning demonstrates the ability of the proposed E-CNNs to increase discriminations between various classes in the histopathological colon images dataset. In comparison to the recent study by *Stoean (2020)*, our ensemble E-CNNs has shown better performance. They constructed CNN from scratch and then used evolutionary algorithms(EA) to fin-tune its parameters. Their classification accuracy on the colon histopathlogical images dataset was 92%. We find that the proposed method's superior due to expanding deeper architecture and utilizing ensemble learning.

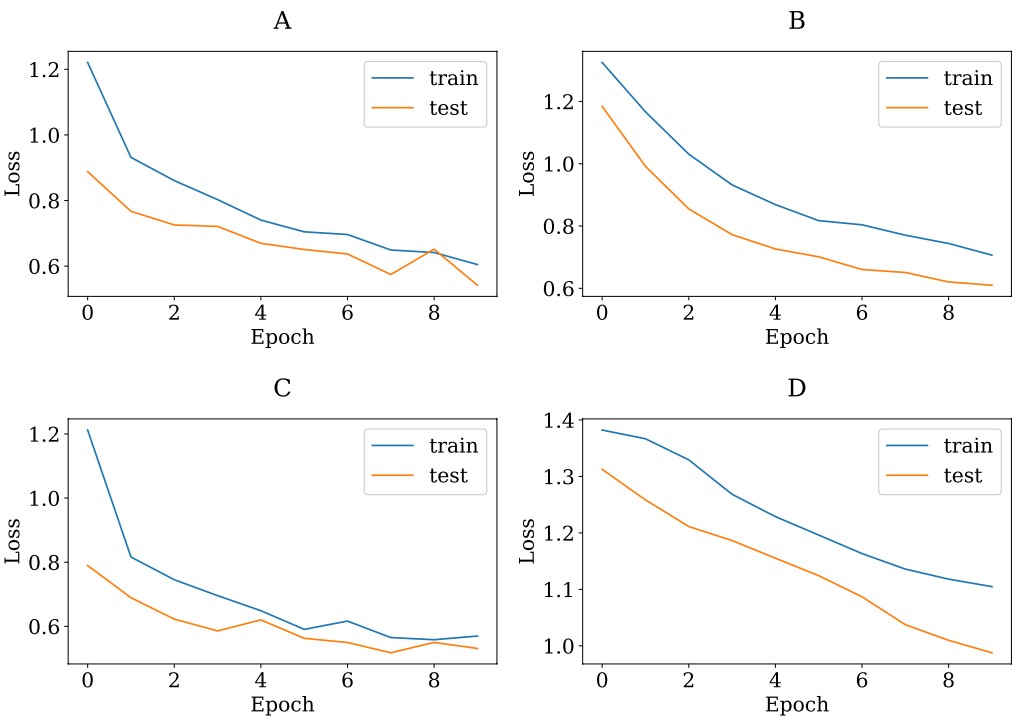

**Figure 9** The loss learning curves of training and testing derived from the four modified CNN base learners, when the number of epochs is ten on the colon histopathological image benchmark Stoean's dataset used in this study: (A) modified DenseNet121, (B) modified InceptionV3, (C) modified MobileNetV2, and (D) modified VGG16.

Moreover, the obtained classification accuracies were compared with the pretrained models GoogleNet and AlexNet in *Popa (2021)*. The proposed method exceeded the pretrained models. The classification accuracy of GoogleNet and AlexNet on colon histopathological images was 85.62% and 89.53%, respectively. The average accuracy rate difference between the proposed method and these pretrained models was more than 10% and 6%, respectively, which was large and statistically significant. Two critical observations are to be made here: first, adapting pretrained models to a specific task increases performance. Second, using pretrained models as a feature extraction without the softmax classifier may degrade the classification accuracy in the colon histopathological image dataset.

To verify that the modified pretrained models are not overfitted, we re-trained them through a number of 30 epochs as in Kather's work (*Kather et al., 2016*). Figures 10 and 11 present the training and validation charts for all the proposed models after being re-trained. According to Fig. 10, validation accuracy, the validation curves for the modified models' graphs increased dramatically after epoch number ten. Actually, they outperformed the training curves. This indicates that the modified models were trained well and avoided the overfitting issue. Figure 11 provides loss curves for the modified models with the 30 epochs It is clear that there is a reduction in the validation loss compared to the training loss, which is noticeable in the delivered loss curves for the individuals of the proposed E-CNN.

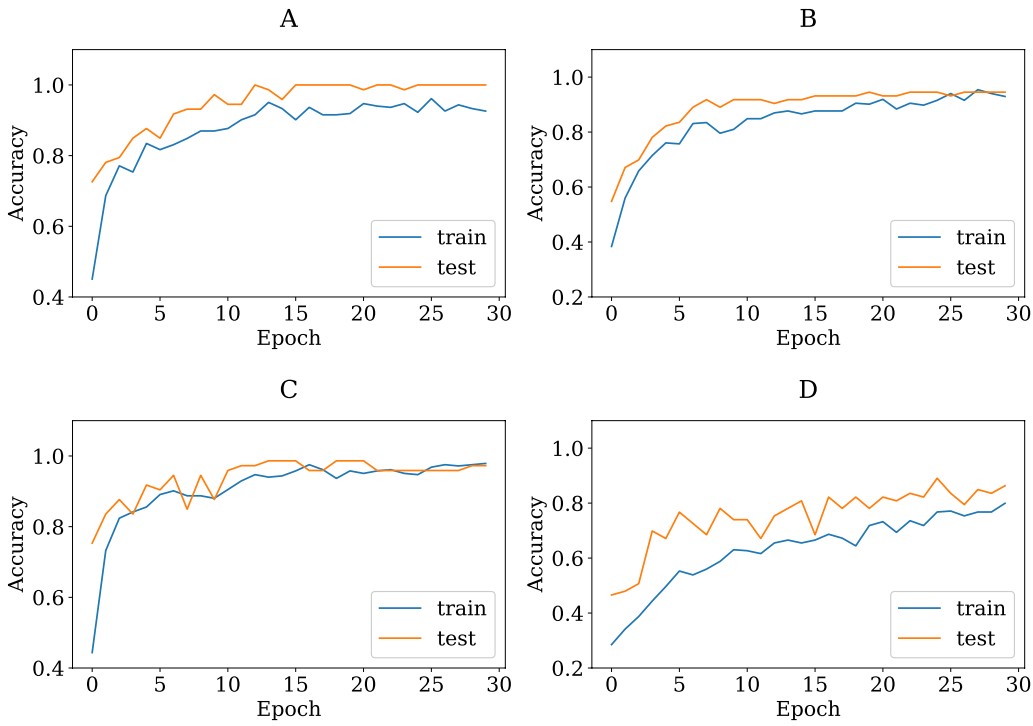

**Figure 10 The accuracy learning curves of training and testing derived from the four modified CNN base learners, when the number of epochs is 30 on the colon histopathological image benchmark Stoean's dataset used in this study: (A) modified DenseNet121, (B) modified InceptionV3, (C) modified MobileNetV2, and (D) modified VGG16.**

The results of the E-CNN and their individuals' base learning with the 30 epochs are illustrated in Table 6. As depicted in this table, the results show that the modified models outperformed the same models when the number of epochs was a set of 10. For example, the modified DenseNet121, MobileNetV2, inceptionv3, and VGG16 with 30 epochs increased the accuracy by 6%, 2%, 4%, and 7%, respectively. This may indicate greater success when increasing the number of epochs and making the deep learning models train enough. Furthermore, the increased performance of the base learner affects the ensemble models. Thus, the results of E-CNN (product rule) and E-CNN (majority voting) are increased by around 2% and 1.0%, respectively compared to the same ensemble when the number of epochs was ten. Figure 12 confirms the confusion matrix of the E-CNN (product rule). These results indicate that the individual learners that we have modified and their ensemble perform robustly better and are not overfitted when increasing the number of epochs.

Moreover, to validate the proposed modified models and their ensembles, we applied these models to the second colon histopathological dataset called the Kather dataset (*Kather et al., 2016*). This dataset contains 5,000 histological images of human colon cancer from eight distinct kinds of tissue. Table 7 gives the accuracy, sensitivity, and specificity of the proposed individual pretrained models, E-CNN (product rule), and E-CNN (majority voting). Besides, the proposed modified models and their ensembles are compared to

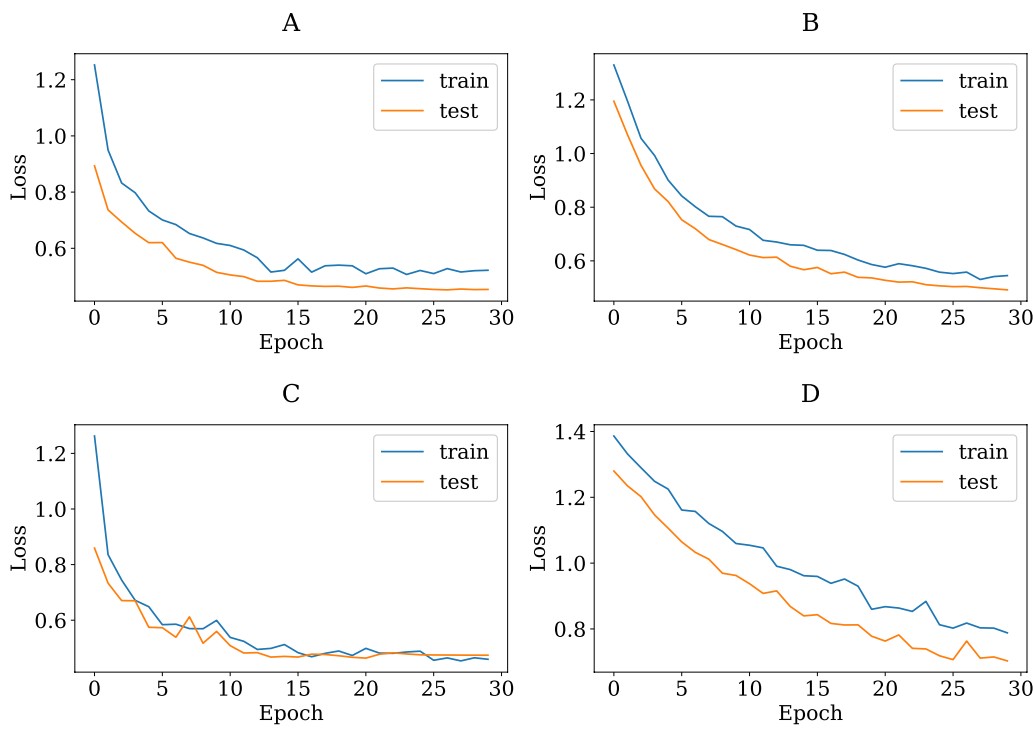

**Figure 11** The loss learning curves of training and testing derived from the four modified CNN base learners, when the number of epochs is 30 on the colon histopathological image benchmark Stoean's dataset used in this study: (A) modified DenseNet121, (B) modified InceptionV3, (C) modified Mo­bileNetV2, and (D) modified VGG16.

similar experiments previously used to assess the classification of the Kather dataset. Based on the results, the proposed modified models are able to separate eight different classes in the histopathological images. Both modified InceptionV3 and DenseNet121 achieved testing accuracy of roughly 89%, with a standard deviation of less than 0.5%. These results outperform the ResNet152 feature extraction results in *Ohata et al. (2021)* by around 9%. That is because the fine-tuned is capable of extracting high-level features from the input images. Furthermore, by using the modified VGG16 on the same dataset, the obtained result has roughly 83% test accuracy, while *Rachapudi & Lavanya Devi (2021)* achieved a test accuracy of 77% when utilizing CNN architecture. This implies that the modification to the pretrained models gets an acceptable result on the histopathological image dataset. The E-CCN (product rule) and E-CNN (majority voting) achieved promising results on the Kather dataset. As shown in Table 7, the E-CCN (product rule) and E-CNN (majority voting) performed better than all individual models, with an accuracy of 91.28% and 90.63%, respectively, which is better than the DensNet121 with only around 2%. These results demonstrate the effectiveness of the proposed modified pretrained models and their ensemble in this classification task.

**Table 7 Evaluation results for the proposed E-CNN, its individuals (modified TL models) when number of epochs = 30, and the standard TL models on Kather's colon histopathlogical images dataset (*Kather et al., 2016*) based on the average accuracy, sensitivity, specificity, and average standard deviation (STD) in 10 runs, best results in bold.**

| Pretrained models | Accuracy | Sensitivity | Specificity |
|---|---|---|---|
| CNN architecture in *Rachapudi & Lavanya Devi (2021)* | 77.0 | – | – |
| ResNet152 feature extraction in *Rachapudi & Lavanya Devi (2021)* | 80.004 ± 1.307 | – | – |
| NASNetMobile feature extraction in *Ohata et al. (2021)* | 89.263 ± 1.704 | – | – |
| Modified DenseNet121 | 89.4. ± 0.56 | 78.32 ± 0.49 | 99.0 ± 0.2 |
| Modified MobileNetV2 | 87.27 ± 0.57 | 76.4 ± 0.5 | 98.7 ± 0.43 |
| Modified InceptionV3 | 89.04 ± 0.36 | 78.0 ± 0.32 | 99.4 ± 0.64 |
| Modified VGG16 | 83.3 ± 1.38 | 72.9 ± 1.26 | 99.1 ± 0.0 |
| **Proposed E-CNN (product)** | **91.28 ± 3.4** | **79.97 ± 3.0** | **99.1 ± 0.0** |
| Proposed E-CNN (Majority voting) | 90.63 ± 4.03 | 79.4 ± 4.02 | 99.1 ± 0.0 |

## Discussion

According to the above experimental results, it is clear that the proposed E-CNNs and adapted TL predictive models outperform other state-of-the-art models and standard pretrained models in the colon histopathological image classification task. The experimental results indicate that adapting the pretrained models for medical image classification tasks improves classification tasks. The results in Tables 4 and 5 demonstrate the critical importance of the introduced adapted models (DenseNet121, MobileNetV2, InceptionV3, and VGG16) in comparison to conventional methods. For example, the adaptive DenseNet model outperformed the standard DenseNet model. These findings show that tailoring the pretrained models to a specific classification task can boost performance. It has also been experimentally verified that using these models in medical image classification results in superior performance when compared to training CNN from scratch (as in previous works by *Postavaru et al., 2017* and *Stoean, 2020*). One reason for this finding is that training a CNN from scratch would necessitate a large number of training samples. Moreover, it must be confirmed that the large number of parameters of the CNN are trained effectively and with a high degree of generalization to obtain acceptable results. Thus, the limitation of the number of training samples causes overfitting in classification tasks. Furthermore, based on the results, it was found that the selection of appropriate hyperparameters in pretrained models plays a vital role in the proper learning and performance of these models.

In this study, two ensemble learning (E-CNN (Majority voting), E-CNN (product rule)) models had been designed to further boost the colon histopathological image classification performance. In the proposed ensemble learning models, the adaptive pretrained models were used as base classifiers. Through the experimental results, one can find that ensemble learning outperformed using individual classifiers. Furthermore, using product rules in the ensemble allows the probabilities of independent events to be fused, ultimately improving performance. This finding is in line with the results in Table 4, where the proposed E-CNN (product) outperformed the proposed E-CNN (majority voting).

Furthermore, the *T*-test is used to compare the proposed E-CNN product to the previously related studies on the same dataset. This test is performed to prove that

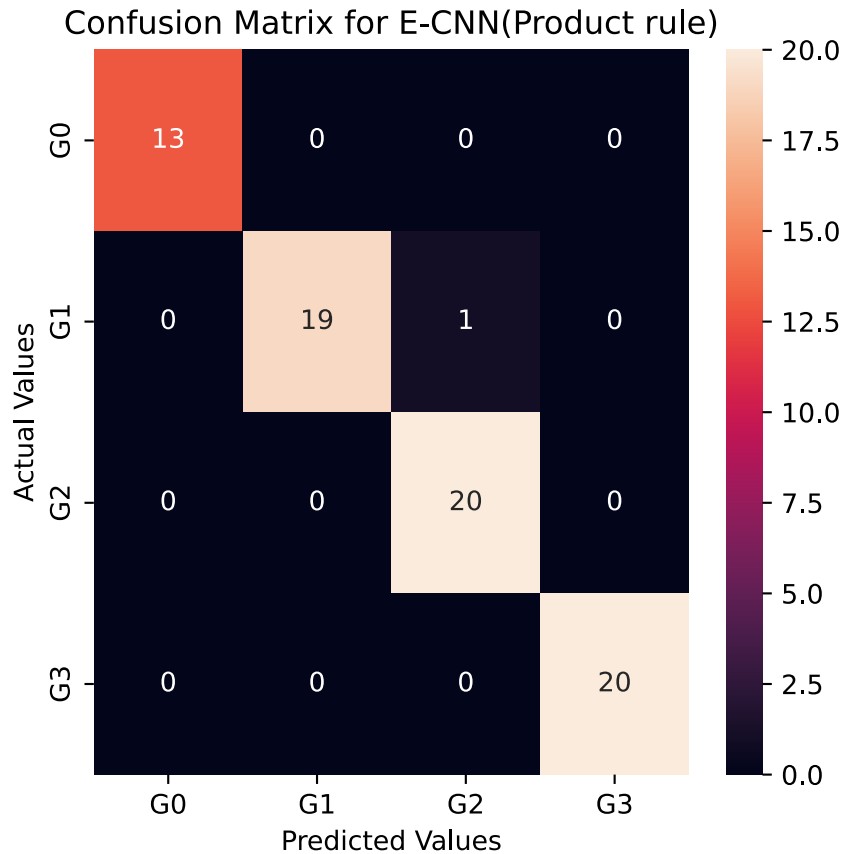

**Figure 12** Confusion matrix of the E-CNN (product rule) on the Stoean testing dataset when number of epochs = 30.

the improvement made by the proposed E-CNN (product) and the state-of-the-art is statistically significant. The $T$-test is carried out based on the average accuracy and the standard deviation of the test samples (20% of the dataset), which are obtained by the E-CNN (product) over ten independent runs. By handling a $T$-test with a 95% spectrum of significance (alpha = 0.05) on the collected $p$-values and the classification accuracy, the corresponding difference statistics are shown in Table 5. As shown in Table 5, the proposed E-CNN (product) outperforms most of the related works on the colon histopathological image dataset, where the majority of the $p$-values of < 0.0001. For example, comparing the proposed E-CCNN with CNN from scratch in *Stoean (2020)*, E-CNN is significantly better with a $p$-value < 0.001. These findings show that using the E-CNN (product) is effective for handling medical image classification tasks.

In summary, it has been demonstrated that the use of the proposed TL models assists in the colon histopathological image classification task, which can be used in the medical domain. Besides, using ensemble learning for the machine learning classification tasks can improve the classification results.

## CONCLUSION AND FUTURE WORK

Deep learning plays a key role in diagnosing colon cancer by grading captured images from colon histopathological images. In this study, we introduced a new set of transfer learning-based methods to help classify colon cancer from histopathological images, which can be used to discriminate between different classes in this domain. To solve this classification task, the pre-trained CNN models DenseNet121, MobileNetV2, InceptionV3, and VGG16 were used as backbone models.

We introduced the TL technique based on a block-wise fine-tuning process to transfer learned experience to colon histopathological images. To accomplish this, we added new dense and drop-out layers to the pretrained models, followed by new FCC with softmax layers to handle the four-class classification task. The adaptability of the proposed models has been enhanced further by the utilized ensemble learning. Two deep ensemble learning methods (E-CNN (product) and E-CNN (Majority voting)) have been proposed. The adapted pretrained models were used as individual classifiers in these proposed ensembles. Next, their output probabilities were fused using the majority voting and the product rule. The acquired results revealed the efficiency of the suggested E-CNNs and their individuals.

We achieved accuracy results of 95.20% and 94.52% for the proposed E-CNN (product) and E-CNN (majority voting), respectively. The proposed E-CNNs and its individual performances were evaluated and compared against the standard (without adaption) pretrained models (DenseNet121, MobileNetV2, InceptionV3, and VGG16 models) as well as state-of-the-art pretrained models and CNN from scratch on colon histopathological images. On all evaluation metrics and the colon histopathological images benchmark dataset, the proposed E-CNNs considerably outperformed the standard pretrained and state-of-the-art CNN from scratch models. The results indicate that the adaptation of pretrained models for TL is a viable option for dealing with the limited number of samples in any new classification task. As a result, the findings indicate that E-CNNs are being used in diagnostic pathology to assist pathologists in making final decisions and accurately diagnosing colon cancer.

Future research could be considered to introduce a new strategy to select the best hyperparameters for the adaptive pretrained models—we recommend wrapper methods for this task.

## ACKNOWLEDGEMENTS

The author would like to thank Dr. Ruba Abu Khurmaa for her assistance in drawing the figures and reviewing the whole manuscript.

### Funding

This research was financed by a grant from the Deanship of Scientific Research and Innovation at Al-Balqa Applied University (Al-Salt-Jordan) for supporting this study (Grant agreement number: DSR-2020#335). The funders had no role in study design, data collection and analysis, decision to publish, or preparation of the manuscript.

### Grant Disclosures

The following grant information was disclosed by the author:

The Deanship of Scientific Research and Innovation at Al-Balqa Applied University (Al-Salt-Jordan): DSR-2020#335.

### Competing Interests

The authors declare there are no competing interests.

### Author Contributions

- Dheeb Albashish conceived and designed the experiments, performed the experiments, analyzed the data, performed the computation work, prepared figures and/or tables, authored or reviewed drafts of the article, and approved the final draft.

### Data Availability

Third-party data is available at

https://sites.google.com/site/imediatreat/

The code is available at:

https://colab.research.google.com/drive/19-S9l2XN6wTmqvi-navTGq6KFnadQFGX?usp=sharing.

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
