# Peer review of "Ensemble of adapted convolutional neural networks (CNN) methods for classifying colon histopathological images"

_PeerJ Computer Science, doi:10.7717/peerj-cs.1031_

## Round 0.1 · original submission · Major Revisions

In particular, the experimental results and design should be discussed in detail and sufficiently discussed.

·

Basic reporting

Your work is good, but there are many typos. The English of the article should be checked by an expert. I've marked some of the misspellings in the PDF. I added a comment. You must comply with the warnings. The abstract article is a little weak compared to its content. It should summarize the article in full. Shapes must be developed. All suggested citations should be included. There are too many repeated citations in the article. The article should be rearranged as incoming.

Experimental design

The Experimental Result section of the article is a bit weak compared to the other sections. This section is the heart of the article. This is the part that belongs to you. Therefore, this section should be developed. The resolution of the figures should be increased. The biggest shortcoming is that there is no confusion matrix. There are calculations, but there are no confusion matrices used to make these calculations. Add at least the confusion matrix of the model you propose (the most successful model, E-CNN). Include the ROC curve if possible. These will enable the reader to better understand and evaluate the article.

Validity of the findings

As stated in the previous comments, more detailed analysis is required to prove the validity of the findings.

Additional comments

Correct the article by considering the warnings and comments in the PDF uploaded to the system. Also take into account the comments made in general.

Reviewer 2 ·

Basic reporting

Figures' quality is not enough.
(For more detail, see attachments)

Experimental design

The methods, especially how the ensemble learning was applied in this study, was not explained in detail.
(For more detail, see attachments)

Validity of the findings

The experimental results (training length of networks) is not enough to compare methods.
It was not provided how many samples were used to compute the statistical p-values.
(For more detail, see attachments)

Additional comments

The motivation of this study is promising. However, it is not easy to make sure of the superiority of the proposed model with such short training.

Annotated reviews are not available for download in order to protect the identity of reviewers who chose to remain anonymous.

Reviewer 3 ·

Basic reporting

The paper has some drawbacks:
1. The overview of the related papers should be expanded. It's a very popular research area. I suggest the authors evaluate these papers related to Transfer Learning and Ensemble Learning Techniques:
https://www.nature.com/articles/s41598-021-93783-8
https://ieeexplore.ieee.org/document/9107128
2. The presentation of Figures 7, 8 should be improved.

Experimental design

What image preprocessing technique do the author use?
The authors should present confusion matrices for each dataset.

Validity of the findings

The authors used only one dataset for testing and validation. It's not enough.

---

## Round 0.2 · Minor Revisions

It is appropriate to accept this article after the plots in the Figures are labelled.

·

Basic reporting

All requested corrections have been carefully made. The article can be published in my opinion.

Experimental design

All requested corrections have been carefully made. The article can be published in my opinion.

Validity of the findings

All requested corrections have been carefully made. The article can be published in my opinion.

Additional comments

All requested corrections have been carefully made. The article can be published in my opinion.

Reviewer 2 ·

Basic reporting

The manuscript meets this journal's standards

Experimental design

The manuscript meets this journal's standards

Validity of the findings

The manuscript meets this journal's standards

Reviewer 3 ·

Basic reporting

The authors addressed all my comments. The paper can be accepted in the present form.

Experimental design

-

Validity of the findings

-

Additional comments

-

---

## Round 0.3 · accepted · Accept

The required revisions have been made.